# A Bayes–Sard Cubature Method

**Toni Karvonen**
Aalto University, Finland

toni.karvonen@aalto.fi

**Chris. J. Oates**
Newcastle University, UK
Alan Turing Institute, UK
chris.oates@ncl.ac.uk

**Simo Särkkä**
Aalto University, Finland

simo.sarkka@aalto.fi

## Abstract

This paper focusses on the formulation of numerical integration as an inferential task. To date, research effort has largely focussed on the development of Bayesian cubature, whose distributional output provides uncertainty quantification for the integral. However, the point estimators associated to Bayesian cubature can be inaccurate and acutely sensitive to the prior when the domain is high-dimensional. To address these drawbacks we introduce Bayes–Sard cubature, a probabilistic framework that combines the flexibility of Bayesian cubature with the robustness of classical cubatures which are well-established. This is achieved by considering a Gaussian process model for the integrand whose mean is a parametric regression model, with an improper prior on each regression coefficient. The features in the regression model consist of test functions which are guaranteed to be exactly integrated, with remaining degrees of freedom afforded to the non-parametric part. The asymptotic convergence of the Bayes–Sard cubature method is established and the theoretical results are numerically verified. In particular, we report two orders of magnitude reduction in error compared to Bayesian cubature in the context of a high-dimensional financial integral.

## 1 Introduction

This paper considers the numerical approximation of an integral $I(f^\dagger) \coloneqq \int_D f^\dagger \mathrm{d}\nu$ of a continuous integrand $f^\dagger : D \to \mathbb{R}$ against a Borel distribution $\nu$ defined on a domain $D \subseteq \mathbb{R}^d$. The approximation of such integrals is a fundamental task in applied mathematics, statistics and machine learning, and is usually achieved using an $n$-point *cubature rule*

$$I_n(f^\dagger) \coloneqq \sum_{i=1}^n w_i f^\dagger(x_i) \approx I(f^\dagger)$$

with some *weights* $w = (w_1, \ldots, w_n) \in \mathbb{R}^n$ and *points* (or *nodes*) $X = \{x_1, \ldots, x_n\} \subset \mathbb{R}^d$. The scope and ambition of modern scientific and industrial computer codes is such that the integrand $f^\dagger$ can often represent the output of a complex computational model. In such cases the evaluation of the integrand is associated with a substantial resource cost and, as a consequence, the total number of evaluations will be limited. The research challenge, in these circumstances, manifests not merely in the design of a cubature method but also in the assessment of the associated error.

The (generalised) *Bayesian cubature* (BC) method [27, 35, 29] provides a statistical approach to error assessment. In brief, let $\Omega$ be a probability space and consider a hypothetical Bayesian agent who represents their epistemic uncertainties in the form of a stochastic process $f : D \times \Omega \to \mathbb{R}$. This stochastic process must arise from a Bayesian regression model and be consistent with obtained evaluations of the true integrand, typically provided on a discrete point set $\{x_i\}_{i=1}^n \subset D$; that is $f(x_i, \omega) = f^\dagger(x_i)$ for almost all $\omega \in \Omega$. The stochastic process acts as a stochastic model for the

integrand $f^\dagger$, implying a random variable $\omega \mapsto \int_D f(\cdot, \omega) \mathrm{d}\nu$ that represents the agent's epistemic uncertainty for the value of the integral $I(f^\dagger)$ of interest.

The output of a (generalised) BC method is the law of the random variable $\omega \mapsto \int_D f(\cdot, \omega) \mathrm{d}\nu$. The mean of this output provides a point estimate for the integral, whilst the standard deviation indicates the extent of the agent's uncertainty regarding the integral. The properties of this probabilistic output have been explored in detail for the case of a centred Gaussian stochastic process (the *standard* BC method): In certain situations the mean has been shown to coincide with a kernel-based integration method [32] that is rate-optimal [1, 5], robust to misspecification of the agent's belief [19, 20] and efficiently computable [32, 22, 23, 18]. The non-Gaussian case and related extensions have been explored empirically in [24, 37, 13, 31, 6]. The method has also been discussed in connection with *probabilistic numerics*; see [10, 14, 7] for general background.

However, it remains the case that non-probabilistic numerical integration methods, such as Gaussian cubatures [11] and quasi-Monte Carlo methods [16], are more widely used, due in part to how their ease-of-use or reliability are perceived. This is despite the well-known fact that the trapezoidal rule and other higher-order spline methods [8] can be naturally cast as Bayesian cubatures if the stochastic process $f$ is selected suitably [10]. It is also known that Gaussian cubature can be viewed as a special (in fact, degenerate) case of a kernel method [46, 21]. However, no overall framework to derive probabilistic analogies of popular cubatures, with corresponding ease-of-use and reliability, has yet been developed.

This paper argues that the perceived performance gap between probabilistic and non-probabilistic methods should be reconsidered. To this end, we consider a non-parametric Bayesian regression model augmented with a parametric component. The features in the parametric component, that is the pre-specified finite set of basis functions, will be denoted $\pi$. Then, an improper prior limit on the regression coefficients (see [48, 33] and [42, Sec. 2.7]) is studied. This gives rise to *Bayes–Sard cubature*[1] (BSC), which differs at a fundamental level to standard BC, in that the functions in $\pi$ are now exactly integrated. The extension is similar, though not identical, to that proposed in 1974 by Larkin [28] and in 1991 by O'Hagan [35], and non-probabilistic versions have appeared independently in [3, 9] in the context of interpolation with conditionally positive definite kernels and optimal approximation in reproducing kernel Hilbert spaces. For other recent work, see also [40, Sec. 2.4]. Our contributions therefore include (i) establishing a coherent and comprehensive Gaussian process framework for BSC; (ii) rigorous study of convergence and conditions that need to be established on $\pi$; (iii) empirical experiments that demonstrate improved accuracy in high dimensions and robustness to misspecified kernel parameters compared to BC; and (iv) the important observation that, when the dimension of the function space $\pi$ matches the number of cubature nodes, the BSC method can be used to endow *any* cubature rule with a meaningful probabilistic output.

## 2  Methods

This section contains our novel methodological development, which begins with specifying a Bayesian regression model for the integrand.

### 2.1  A Bayesian Regression Model

This section serves to set up a generic Bayesian regression framework, which is essentially identical to that described in [33, 48] and [44, Sec. 4.1.2]. See also [42, Sec. 2.7] and [30]. This will act as the stochastic model $f : D \times \Omega \to \mathbb{R}$ for our subsequent development.

#### 2.1.1  Gaussian Process Prior

Recall that a *Gaussian process* is a function-valued random variable $\omega \mapsto f(\cdot, \omega)$ such that $f(\cdot, \omega) \in C^0(D)$ and $\omega \mapsto Lf(\cdot, \omega)$ is a (univariate) Gaussian for all continuous linear functionals $L$ on $C^0(D)$. Here $\omega$ denotes a generic element of an underlying probability space $\Omega$. See [4]

for further background. Following the notational convention in [42], we suppress the argument $\omega$ and denote by $f(x) \sim \mathcal{GP}(s(x), k(x, x'))$ a Gaussian process with mean function $s \in C^0(D)$ and positive definite covariance kernel $k \in C^0(D \times D)$. The characterising property of this Gaussian process is that $f(x_1), \ldots, f(x_n)$ are jointly Gaussian with the mean vector $[s(x_1), \ldots, s(x_n)]$ and covariance matrix $[K_X]_{ij} = k(x_i, x_j)$ for all sets $X = \{x_1, \ldots, x_n\} \subset D$.

Our starting point in this paper will be to endow a hypothetical Bayesian agent with the following prior model for the integrand:

**Definition 2.1** (Prior). Let $\pi$ be a finite-dimensional linear subspace of real-valued functions on $D$ and $\{p_1, \ldots, p_Q\}$ a basis of $\pi$, so that $Q = \dim(\pi)$. Then, for some positive definite covariance matrix $\Sigma \in \mathbb{R}^{Q \times Q}$, we consider the following hierarchical *prior* model:

$$f(x) \mid \gamma \sim \mathcal{GP}\big(s(x), k(x, x')\big), \quad s(x) = \sum_{j=1}^{Q} \gamma_j p_j(x), \quad \gamma \sim \mathcal{N}(0, \Sigma).$$

The mean function $s \in \pi$ is parametrised by $\gamma_1, \ldots, \gamma_Q \in \mathbb{R}$. Such a prior could arise, for example, when a parametric linear regression model is assumed and a non-parametric discrepancy term added to allow for misspecification of the parametric part [25]. Note that a non-zero mean $\eta \in \mathbb{R}^Q$ could be specified for $\gamma$; this is done in the derivations contained in supplementary material. For the proposed method to be implementable, the functions $p_1, \ldots, p_Q$ must be known and their integrals available.

### 2.1.2 Gaussian Process Posterior

In a regression context, the data consist of input-output pairs $\mathcal{D}_X = \{(x_i, f^\dagger(x_i))\}_{i=1}^n$, based on a finite point set $X$ that, in this paper, is considered fixed. The elements of $X$ are assumed to be distinct. Our interest is in the Bayesian agent's belief, after the data $\mathcal{D}_X$ are observed. The *posterior* is defined as the law of the stochastic process which is obtained by conditioning the prior stochastic process on $\mathcal{D}_X$. That the posterior, denoted $f \mid \mathcal{D}_X$, is again a Gaussian stochastic process is a well-known result (for technical details, see e.g. [38]).

Let $f_X$ (resp. $f_X^\dagger$) denote the column vector with entries $f(x_i)$ (resp. $f^\dagger(x_i)$). Let $p(x)$ be the row vector with entries $p_j(x)$ and let $P_X$ denote the $n \times Q$ *Vandermonde matrix* with $[P_X]_{i,j} = p_j(x_i)$. Let $k_X(x)$ denote the row vector with entries $k(x, x_j)$ and let $K_X$ denote the *kernel matrix* with $[K_X]_{i,j} = k(x_i, x_j)$. For the prior in Def. 2.1 we have the following result:

**Theorem 2.2** (Posterior). *In the posterior,* $f(x) \mid \mathcal{D}_X \sim \mathcal{GP}(s_{X,\Sigma}(f^\dagger)(x), k_{X,\Sigma}(x, x'))$ *where*

$$\begin{aligned} s_{X,\Sigma}(f^\dagger)(x) &= k_X(x)\alpha + p(x)\beta \\ &= [k_X(x) + p(x)\Sigma P_X^\top][K_X + P_X \Sigma P_X^\top]^{-1} f_X^\dagger, \end{aligned} \quad (1)$$

$$\begin{aligned} k_{X,\Sigma}(x, x') &= k(x, x') + p(x)\Sigma p(x')^\top \\ &\quad - [k_X(x) + p(x)\Sigma P_X^\top][K_X + P_X \Sigma P_X^\top]^{-1}[k_X(x') + p(x')\Sigma P_X^\top]^\top \end{aligned} \quad (2)$$

*and the coefficients $\alpha$ and $\beta$ are defined via the invertible linear system*

$$\begin{bmatrix} K_X & P_X \\ P_X^\top & -\Sigma^{-1} \end{bmatrix} \begin{bmatrix} \alpha \\ \beta \end{bmatrix} = \begin{bmatrix} f_X^\dagger \\ 0 \end{bmatrix}. \quad (3)$$

The proofs for all results are contained in the supplementary material, unless otherwise stated. Note that the posterior is consistent with the data $\mathcal{D}_X$, in the sense that the posterior mean $s_{X,\Sigma}(f^\dagger)(x)$ coincides with the value $f^\dagger(x)$ at the locations $x \in X$ and, moreover, the posterior variance vanishes at each $x \in X$. These facts imply that sample paths from $f \mid \mathcal{D}_X$ almost surely satisfy $f_X = f_X^\dagger$.

**Remark 2.3** (Standard Bayesian cubature; BC). Based on Eqns. 1 and 2, it is apparent that if we set $\pi = \emptyset$, then the posterior reduces to a Gaussian process with mean and covariance

$$s_{X,0}(f^\dagger)(x) = k_X(x)K_X^{-1}f_X^\dagger, \quad k_{X,0}(x, x') = k(x, x') - k_X(x)K_X^{-1}k_X(x')^\top.$$

This is precisely the stochastic process used in the standard BC method [29, 5].

The need for the Bayesian agent to elicit a covariance matrix $\Sigma$ appears to prevent automatic use of this prior model. For this reason, we consider the *flat prior limit* as $\Sigma^{-1} \to 0$, which corresponds to a particular encoding of an absence of prior information about the value of the parameter $\gamma$ in Def. 2.1.

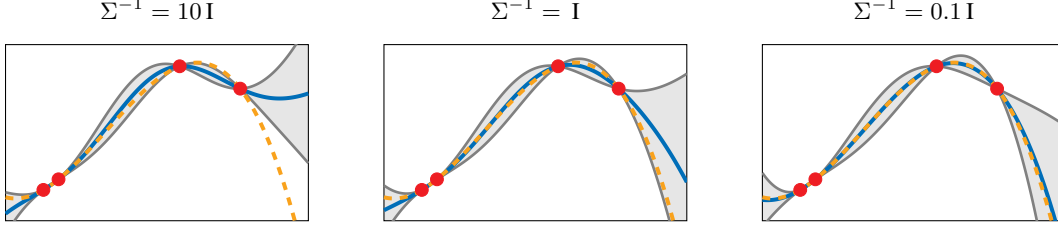

<div style="text-align:center">

$\Sigma^{-1} = 10\,I$      $\Sigma^{-1} = I$      $\Sigma^{-1} = 0.1\,I$

</div>

Figure 1: Posterior mean (blue) and 95% credible intervals (gray) given four data points (red) for the prior model of Def. 2.1, with the linear space $\pi$ taken as the set of polynomials with degree $\leq 3$. The Gaussian kernel with length-scale $\ell = 0.8$ was used. The unique polynomial interpolant of degree 3 to the data (dashed) is plotted for comparison. Note convergence of the posterior mean to the polynomial interpolant as $\Sigma^{-1} \to 0$.

### 2.1.3 Flat Prior Limit

In this section we ask whether the Gaussian process posterior is well-defined in the flat prior limit $\Sigma^{-1} \to 0$. For this, we need the concept of unisolvency [49, Sec. 2.2]:

**Definition 2.4** (Unisolvency). Let $\pi$ be a finite-dimensional linear subspace of real-valued functions on $D$. A set $X = \{x_1, \ldots, x_n\} \subset D$ with $n \geq \dim(\pi)$ is called $\pi$-*unisolvent* if the zero function is the only element in $\pi$ that vanishes on $X$. (Examples are provided in Sec. B of the supplement.)

**Theorem 2.5** (Flat prior limit). *Assume that $X$ is a $\pi$-unisolvent set. For the prior in Def. 2.1 we have that $s_{X,\Sigma}(f^\dagger) \to s_X(f^\dagger)$ and $k_{X,\Sigma} \to k_X$ pointwise as $\Sigma^{-1} \to 0$, where*

$$s_X(f^\dagger)(x) = k_X(x)\alpha + p(x)\beta, \tag{4}$$

$$\begin{aligned} k_X(x, x') &= k(x, x') - k_X(x) K_X^{-1} k_X(x')^\top \\ &\quad + \big[k_X(x) K_X^{-1} P_X - p(x)\big] \big[P_X^\top K_X^{-1} P_X\big]^{-1} \big[k_X(x') K_X^{-1} P_X - p(x')\big]^\top, \end{aligned} \tag{5}$$

*and the coefficients $\alpha$ and $\beta$ are defined via the invertible linear system*

$$\begin{bmatrix} K_X & P_X \\ P_X^\top & 0 \end{bmatrix} \begin{bmatrix} \alpha \\ \beta \end{bmatrix} = \begin{bmatrix} f_X^\dagger \\ 0 \end{bmatrix}. \tag{6}$$

The following observation, illustrated in Fig. 1, will be important:

**Proposition 2.6.** *Assume that $X$ is a $\pi$-unisolvent set. Then $s_X(p) = p$ whenever $p \in \pi$.*

*Proof.* If $p \in \pi$, there exist coefficients $\beta_1', \ldots, \beta_Q'$ such that $p = \sum_{i=1}^Q \beta_j' p_j$. That is, a particular solution of Eqn. 6 is $\alpha = 0$ and $\beta = \beta'$. The linear system being invertible, this must be the only solution. We deduce that $s_X(p) = p$. $\square$

In particular, if $\dim(\pi) = n$, the posterior mean reduces to the unique interpolant in $\pi$ to the data $\mathcal{D}_X$ while the posterior covariance is non-zero. This observation will enable us to endow any classical cubature rule with a non-degenerate probabilistic output in Sec. 2.4. Next we turn our attention to estimation of the unknown value of the integral.

### 2.2 The Bayes–Sard Framework

Recall that the output of a generalised BC method is the push-forward $\omega \mapsto \int_D f(\cdot, \omega)\,d\nu$ of the stochastic process $f \mid \mathcal{D}_X$ through the integration operator $I$. This random variable will be denoted $I(f) \mid \mathcal{D}_X$. In this section we present the BSC method, which is based on the prior model with $\Sigma^{-1} \to 0$ studied in Sec. 2.1.3. It will be demonstrated that BSC differs, at a fundamental level, from the standard BC method in that the elements of $\pi$ are exactly integrated.

Let $k_\nu(x) = I(k(\cdot, x))$ denote the *kernel mean* function and $k_{\nu,\nu} = I(k_\nu)$ its integral. Define the row vectors $p_\nu$ and $k_{\nu,X}$ to have respective entries $[p_\nu]_j = I(p_j)$ and $[k_{\nu,X}]_j = k_\nu(x_j)$.

**Theorem 2.7** (Bayes–Sard cubature; BSC). *Consider the Gaussian process*

$$f(x) \mid \mathcal{D}_X \sim \mathcal{GP}\big(s_{X,\Sigma}(f^\dagger)(x), k_{X,\Sigma}(x, x')\big)$$

*defined in Thm. 2.2 and suppose that $X$ is a $\pi$-unisolvent point set. Then, as $\Sigma^{-1} \to 0$, the mean and variance of $I(f) \mid \mathcal{D}_X$ converge to*

$$\mu_X(f^\dagger) = w_k^\top f_X^\dagger \quad \text{and} \quad \sigma_X^2 = k_{\nu,\nu} - k_{\nu,X} K_X^{-1} k_{\nu,X}^\top + \big(k_{\nu,X} K_X^{-1} P_X - p_\nu\big) w_\pi,$$

*respectively, where the weight vectors $w_k \in \mathbb{R}^n$ and $w_\pi \in \mathbb{R}^Q$ are obtained from the solution of the invertible linear system*

$$\begin{bmatrix} K_X & P_X \\ P_X^\top & 0 \end{bmatrix} \begin{bmatrix} w_k \\ w_\pi \end{bmatrix} = \begin{bmatrix} k_{\nu,X}^\top \\ p_\nu^\top \end{bmatrix}. \tag{7}$$

The posterior mean indeed takes the form of a cubature rule, with weights $w_{k,i}$ and points $x_i \in X$. This provides a point estimator for the integral $I(f^\dagger)$, while the posterior variance enables uncertainty to be assessed. The *Bayes–Sard* nomenclature derives from the fact that the associated cubature rule $\mu_X$ is *exact* on the space $\pi$ (recall Prop. 2.6; the proof is also similar):

**Proposition 2.8.** *Assume that $X$ is a $\pi$-unisolvent set. Then $\mu_X(p) = I(p)$ whenever $p \in \pi$.*

Thus we have a probabilistic framework that combines the flexibility of BC with the robustness of classical numerical integration techniques, for instance based on a polynomial exactness criteria being satisfied.

**Remark 2.9.** Rates of convergence identical to those appearing in [5, 20] for the BC can be derived for the BSC method by using results in [49]. Details are contained in Sec. C of the supplement.

### 2.3 Normalised Bayesian Cubature

The difference between BSC and BC is perhaps best illustrated in the case $\pi = \{1\}$, also considered in [24, 41, 18], where constant functions are exactly integrated in BSC but not in BC. Indeed, $P_X = \mathbb{1}$, the $n$-vector of ones, and

$$w_k = \left(I - \frac{K_X^{-1} \mathbb{1} \mathbb{1}^\top}{\mathbb{1}^\top K_X^{-1} \mathbb{1}}\right) K_X^{-1} k_{\nu,X}^\top + \frac{K_X^{-1} \mathbb{1}}{\mathbb{1}^\top K_X^{-1} \mathbb{1}}.$$

These weights have the desirable property of summing up to one; we might therefore call this a *normalised Bayesian cubature* method. Furthermore, if the kernel is parametrised by a length-scale parameter and this parameter is too small, then $w_{k,i} \approx 1/n$, which is a reasonable default. This should be contrasted with BC, for which the weights $w_{k,i} \approx 0$ become degenerate instead.

### 2.4 Reproduction of Classical Cubature Rules

In this section we indicate how *any* cubature rule can be endowed with a probabilistic interpretation under the Bayes–Sard framework. Recall that every continuous positive definite kernel $k$ induces a unique *reproducing kernel Hilbert space* (RKHS) $H(k) \subset C^0(D)$ with norm denoted $\|\cdot\|_k$ [2]. It is well-known that the weights $w_{\text{BC}} := K_X^{-1} k_{\nu,X}^\top \in \mathbb{R}^n$ of the standard BC method (recall Rmk. 2.3) are *worst-case optimal* in $H(k)$:

$$w_{\text{BC}} = \arg\min_{w \in \mathbb{R}^n} e_k(X, w), \qquad e_k(X, w) := \sup_{\|h\|_k \le 1} \left| \int_D h \mathrm{d}\nu - \sum_{i=1}^n w_i h(x_i) \right|,$$

where $e_k(X, w)$ is the *worst-case error* (WCE) of the cubature rule specified by the points $X$ and weights $w$. Furthermore, the posterior standard deviation coincides with $e_k(X, w_{\text{BC}})$. See [26, 43, 32] for details on optimal cubature rules in RKHS. It is now shown that, when $\dim(\pi) = n$, the BSC weights in Thm. 2.7 do not depend on the kernel and the standard deviation coincides with the WCE:

**Theorem 2.10.** *Suppose that $\dim(\pi) = n$ and let $X$ be a $\pi$-unisolvent set. Then*

$$\mu_X(f^\dagger) = w_k^\top f_X^\dagger, \quad w_k^\top = p_\nu P_X^{-1} \quad \text{and} \quad \mu_X(p) = I(p) \quad \text{for every } p \in \pi$$

*and*

$$\sigma_X^2 = e_k(X, w_k)^2 = k_{\nu,\nu} - 2k_{X,\nu} w_k + w_k^\top K_X w_k.$$

*That is, the BSC weights $w_k$ are the unique weights for a cubature rule with the points $X$ such that every function in $\pi$ is integrated exactly and the posterior standard deviation $\sigma_X$ coincides with the WCE in the RKHS $H(k)$.*

**Corollary 2.11.** *Consider an $n$-point cubature rule with points $X$ and non-zero weights $w \in \mathbb{R}^n$ and assume that $\nu$ admits a positive density function (w.r.t. the Lebesgue measure). Then there is a function space $\pi$ of dimension $n$, such that the BSC method recovers $w_k = w$ and $\sigma_X^2 = e_k(X, w)^2$, as defined in Thm. 2.7.*

Thus *any* cubature rule can be recovered as a posterior mean for some prior (briefly alluded to in [35, Sec. 2.3] in a more limited setting and lacking RKHS machinery). Our result goes beyond earlier work in [46, 21], in the sense that the associated posterior is non-degenerate (i.e. has non-zero variance) in the Bayes–Sard framework. Further discussion is provided in Sec. D of the supplement. From a practical perspective, this enables us to simultaneously achieve the same reliable integration performance as popular non-probabilistic rules (see Sec. C.2 in the supplement) *and* to perform formal uncertainty quantification for the integral.

**Remark 2.12.** The function space alluded to in Cor. 2.11 can be constructed explicitly. The general construction is somewhat artificial, but can be made more appealing if the weights arise from, for example, a natural polynomial exactness criterion. See Sec. A.2 of the supplement for details.

## 2.5 On Weakly-Informative Priors

As mentioned earlier, methods similar to ours were proposed by O'Hagan [35]. See also [28, 36, 24] and, in particular, [34, Sec. 3.6]. Following [35], let $k(x, x') = \lambda k_0(x, x')$ for some base kernel $k_0$ and consider the improper prior $p(\gamma, \lambda) \propto 1/\lambda$. It can then be shown that the marginal posterior for $I(f)$ is Student-$t$, with $n - Q$ degrees of freedom and mean equal to $\mu_X(f^\dagger)$ in our work, but whose variance is instead

$$\frac{1}{n - Q - 2}(f_X^\dagger)^\top \big( K_X^{-1} - K_X^{-1} P_X [P_X^\top K_X^{-1} P_X]^{-1} P_X^\top K_X^{-1} \big) f_X^\dagger \times \sigma_X^2.$$

That is, when $n - 2 < Q \leq n$, this posterior is not well-defined. As a consequence of this prior specification one cannot, as opposed to Cor. 2.11 that requires $Q = n$, associate every cubature rule with a non-degenerate posterior. Thus one of the principal advantages of using the weakly-informative informative prior, obtained as a limit of Gaussians, considered in this paper is that the worst-case error can be reinterpreted as a posterior standard deviation. However, to ensure this variance provides meaningful quantification of uncertainty can be challenging. This is discussed in Secs. 3.1 and 3.4.

# 3 Experimental Results

This section contains three numerical experiments, which investigate the empirical performance of the BSC method and the associated uncertainty quantification that is provided. The examples demonstrate that BSC is typically at least as accurate as BC whilst being less sensitive to misspecification of the kernel length-scale parameter.

## 3.1 On Choosing the Kernel Parameters

The stationary kernels often used in Gaussian process regression are parametrised by positive *length-scale*[2] $\ell$ and *amplitude* $\lambda$:

$$k(x, x') = k(x - x') = \lambda k_0\big((x - x')/\ell\big)$$

for, in a slight abuse of notation, some base kernel $k_0$. Adapting these parameters in a data-dependent way is an essential prerequisite for meaningful quantification of uncertainty for the integral. After taking the limit $\Sigma^{-1} \to 0$, that yields the BSC, we proceed to set these parameters independently, following the approach suggested in [5, Sec. 4.1], but as if the prior model were

$$f(x) \mid \ell, \lambda \sim \mathcal{GP}\big(0, \lambda k_0((x - x')/\ell)\big).$$

This procedure, though admittedly somewhat unsound, appears to work well in the examples of Secs. 3.2 and 3.4. That is, we (i) assign $\lambda$ the improper prior $p(\lambda) \propto 1/\lambda$ and marginalise over it so that the

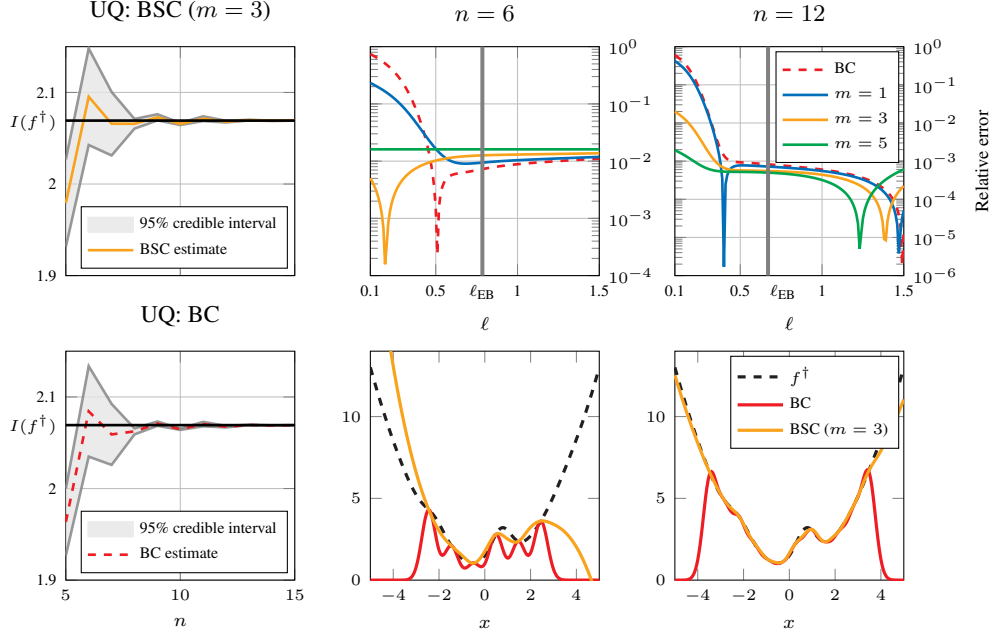

Figure 2: Approximation of the integral in Eqn. 8 using BC and BSC with $\pi = \Pi_m(\mathbb{R})$ for $m = 1, 3, 5$, both based on the Gaussian kernel. The $n$ nodes were placed uniformly on $[-\sqrt{n}, \sqrt{n}\,]$. *Left*: Uncertainty quantification (UQ) provided by BSC with $m = 3$ and BC when kernel parameters were selected as outlined in Sec. 3.1. *Middle & right*: Effect of $\ell$ on approximation accuracy. The upper row presents the relative integration error $|I(f^\dagger) - I_n(f^\dagger)|/I(f^\dagger)$ for each cubature rule $I_n$ as a function of $\ell$. The "optimal" length-scales $\ell_{\text{EB}}$, as computed by EB, are also shown. The lower row contains the corresponding posterior means when an inappropriate value, $\ell = 0.3$, is used. Since $\dim(\Pi_5(\mathbb{R})) = 6$, that BSC for $m = 5$ and $n = 6$ is independent of $\ell$ is a consequence of Thm. 2.10.

BSC posterior becomes Student-$t$ with the mean $\mu_X(f^\dagger)$, variance $(n-2)^{-1}(f_X^\dagger)^\top K_X^{-1} f_X^\dagger \times \sigma_X^2$ and $n$ degrees of freedom [35, Sec. 2.2] and (ii) set $\ell$ using empirical Bayes (EB) based on the Gaussian log-marginal likelihood [42, Sec. 5.4.1]

$$\ell_{\text{EB}} = \underset{\ell > 0}{\arg\max} \left[ -\frac{1}{2}(f_X^\dagger)^\top K_X^{-1} f_X^\dagger - \frac{1}{2}\log\det(K_X) \right].$$

There are of course other possibilities that could be explored, such as cross-validation or, when $Q < n$, using the likelihood of the regression model set up in Sec. 2.1 (see [42, Eqn. 2.45]).

## 3.2 A One-Dimensional Toy Example

Our first example is one-dimensional. The test function and its integral that we considered were

$$f^\dagger(x) = \exp\left(\sin(2x) - \frac{x^2}{5}\right) + \frac{x^2}{2} \quad \text{and} \quad I(f^\dagger) = \frac{1}{\sqrt{2\pi}} \int_{\mathbb{R}} f^\dagger(x) e^{-x^2/2} dx \approx 2.0693. \quad (8)$$

The effect of the length-scale $\ell$ of the Gaussian kernel $k(x, x') = \exp(-(x - x')^2/(2\ell^2))$ on the performance of standard BC and BSC of Sec. 2.2, with

$$\pi = \Pi_m(\mathbb{R}) \coloneqq \text{span}\{x^\alpha : \alpha \in \mathbb{N}_0^d, \alpha_1 + \cdots + \alpha_d \leq m\}, \quad \text{where} \quad x^\alpha = x_1^{\alpha_1} \times \cdots x_d^{\alpha_d},$$

for different $m$, was investigated and the quality of the uncertainty quantification was assessed.

Results are depicted in Fig. 2. It can be observed that the BSC is more robust compared to BC when the length-scale is misspecified (in particular, when it is too small). This is because the polynomial part mitigates the tendency of the posterior mean to revert quickly back to zero. For reasonable values of the length-scale, the accuracy of the different methods is comparable. The BSC and BC provide qualitatively similar quantification of uncertainty and both exhibit a degree of over-confidence, as

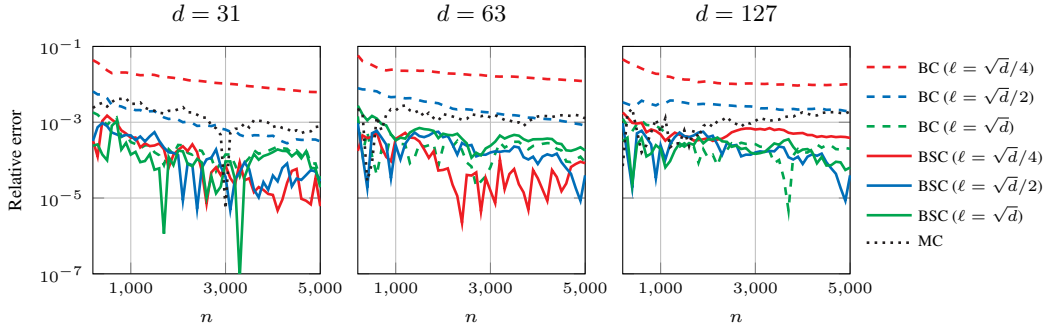

Figure 3: Approximation of the $d$-dimensional integral (9) using BC and BSC with $\pi = \Pi_1(\mathbb{R}^d)$, both based on product Matérn kernel with $\rho = 5/2$ (see Eqn. C15 in the supplement) and length-scale $\ell$. Figures contain relative integration errors for each cubature rule for a given dimension and different length-scales as a function of the number of nodes $n$, drawn randomly from the uniform distribution. Non-monotonicity of the BSC errors is caused by higher accuracy of this method; small fluctuations in error are magnified on the logarithmic scale. Note that the point set is almost surely unisolvent. For comparison, the standard Monte Carlo approximation (MC) is also plotted.

observed already in [5, Sec. 5.1] for the BC and attributed to the manner in which the length-scale is selected. However, BSC is less over-confident. The reason for this is that the BSC variance in Thm. 2.7 is a sum of the BC variance and a positive term.

## 3.3 Zero Coupon Bonds

This section experiments with a high-dimensional zero coupon bond example that has been used previously in numerical experiments for kernel cubature in [22, Sec. 5.5]. See [17, Sec. 6.1] and Sec. E of the supplement for a more detailed account of this experiment. The integral of interest is

$$P(0,T) := \mathbb{E}\left[\exp\left(-\Delta t \sum_{i=0}^{d_T-1} r_{t_i}\right)\right] = \exp(-\Delta t r_{t_0})\mathbb{E}\left[\exp\left(-\Delta t \sum_{i=1}^{d_T-1} r_{t_i}\right)\right], \quad (9)$$

where $r_{t_i}$, $i > 0$, are Gaussian random variables and $r_{t_0}$ is a constant. This $d = d_T - 1$ dimensional integral represents the price at time $t = 0$ of a zero coupon bond with maturity time $T$ and arises from $d_T$-step uniform Euler–Maruyama discretisation of the Vasicek model. Existence of a closed-form solution for $P(0,T)$ makes numerical approximation of Eqn. 9 an attractive high-dimensional benchmark problem.

We transformed the integral (9) onto the hypercube $[0,1]^d$ and compared the accuracy of BC to BSC with $\pi = \Pi_1(\mathbb{R}^d)$. Different dimensions $d$ and length-scales $\ell$ were considered and the product Matérn kernel with smoothness parameter $\rho = 5/2$ (see Eqn. C15 in the supplement) was used. As in Sec. 3.2, it is apparent from Fig. 3 that the BSC is less sensitive to length-scale misspecification compared to the standard BC method. In this misspecified case a two order of magnitude reduction in integration error was observed. This is attributed to the improved extrapolation performance conferred through the polynomial component.

## 3.4 Uncertainty Quantification for Gauss–Patterson Quadrature

In this section we assess the uncertainty quantification provided by Cor. 2.11 for Gauss–Patterson quadrature rules [39], a sequence of nested classical quadrature rules. These rules are nested extensions of the familiar Gaussian quadratures: to an $n$-point quadrature rule $n + 1$ points are added so as to maximise the polynomial degree of the resulting $(2n + 1)$-point quadrature rule and the process is then repeated iteratively. For the uniform measure, these rules have been computed[3] for the sequence $n = (1, 3, 7, 15, 31, 63, 127, 255, 511)$; see [12] for a Gaussian version.

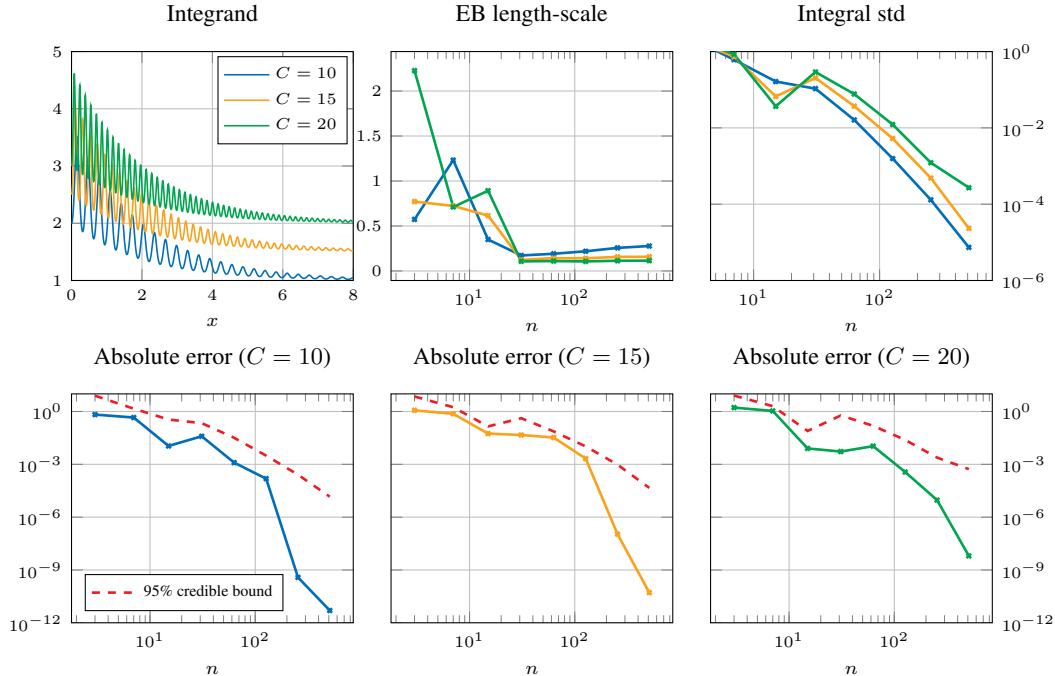

Figure 4: Uncertainty quantification by the BSC for Gauss–Patterson quadrature applied to the integration problem in Eqn. 10. The number of nodes ranges from 3 to 511. The lower row presents the absolute integration error $|I(f_C^\dagger) - I_n(f_C^\dagger)|$ of the $n$-point Gauss–Patterson rule for the three integrals. The plotted credible bound exceeding the integration error indicates that the true integral value is contained within the 95% highest posterior density credible interval.

Our final experiment considered computation of the integrals

$$I(f_C^\dagger) = \frac{1}{8}\int_0^8 f_C^\dagger(x)\mathrm{d}x \quad \text{for} \quad f_C^\dagger(x) = \exp\left(\sin(Cx)^2 - 0.5x\right) + \frac{C}{10}, \ \ C \in \{10, 15, 20\}, \quad (10)$$

which are expected to be more difficult to compute the larger the constant $C$ is (see [5, Sec. 5.1] for a similar example for the standard BC). We again used the Matérn kernel with the smoothness parameter $\rho = 5/2$ and set its length-scale and magnitude parameters for each $n$ as described in Sec. 3.1 The results appear in Fig. 4, where we clearly observe that a larger integral variance is assigned for more difficult integrals and that the true integral value is always contained within the 95% credible interval. In particular, for small $n$ that do not produce useful integral estimates ($n = 3$ yields relative errors between 0.46 and 0.69 and $n = 7$ between 0.31 and 0.44) the posterior variance is large, correctly reflecting significant uncertainty in these estimates. This suggests that the BSC appears to provide sensible uncertainty quantification for Gauss–Patterson rules, at least in this experiment.

## 4   Conclusion

This paper proposed a Bayes–Sard cubature method, which provides an explicit connection between classical cubatures and the Bayesian inferential framework. In particular, we obtained polynomially exact generalisations of standard BC in Thm. 2.7 and demonstrated in Cor. 2.11 how any cubature rule, including widely-used cubature methods, can be recovered as the output of a probabilistic model.

The main practical drawback of standard BC is its acute sensitivity to the choice of kernel. As demonstrated in Sec. 3, the Bayes–Sard point estimator performance is more robust to the choice of kernel and this suggests that fast Gaussian process methods (e.g., [15, 51]) could be used for efficient automatic selection of kernel parameters with little adverse effect on accuracy of the point estimator. On the other hand, further work is required to assess the quality of the uncertainty quantification provided by the BSC method. This will require careful analysis that accounts for how kernel parameters are estimated, and is expected to be technically more challenging (see, e.g., [50]).

**Acknowledgments**

The authors are grateful for discussion with Aretha Teckentrup, Catherine Powell, Fred Hickernell and Filip Tronarp. TK was supported by the Aalto ELEC Doctoral School. CJO was supported by the Lloyd's Register Foundation programme on data-centric engineering. SS was supported by the Academy of Finland projects 266940, 304087 and 313708. This material was based upon work partially supported by the National Science Foundation under Grant DMS-1127914 to the Statistical and Applied Mathematical Sciences Institute. Any opinions, findings, and conclusions or recommendations expressed in this material are those of the author(s) and do not necessarily reflect the views of the National Science Foundation.

## Footnotes

[1]Our terminology is motivated by resemblance to the (non-probabilistic) method of Sard [45] for selecting weights for given $n$ nodes by fixing a polynomial space of degree $m < n$ on which the integration rule must be exact and disposing of the remaining $n - m - 1$ degrees of freedom by minimising an appropriate error functional. See also [47] and [26].

[2]In general, a distinct length-scale parameter for each dimension could be used.

[3]https://people.sc.fsu.edu/~jburkardt/m_src/patterson_rule/patterson_rule.html

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
