[Supplementary Material]

# Supplementary Material

This document contains supplementary material for the article "A Bayes–Sard Cubature Method".

## A  Proof of Results in the Main Text

The prior model (Def. 2.1) used in the main text is

$$f(x) \mid \gamma \sim \mathcal{GP}\big(s(x), k(x, x')\big), \quad s(x) = \sum_{j=1}^{Q} \gamma_j p_j(x), \quad \gamma \sim \mathcal{N}(0, \Sigma).$$

It is straightforward to consider the generalisation $\gamma \sim \mathcal{N}(\eta, \Sigma)$ for potentially non-zero vector $\eta \in \mathbb{R}^Q$; we do this in this supplement.

### A.1  Results on the Regression Model

**Theorem 2.2** (Posterior). *In the posterior, $f(x) \mid \mathcal{D}_X \sim \mathcal{GP}(s_{X,\Sigma}(f^\dagger)(x), k_{X,\Sigma}(x, x'))$ where*

$$\begin{aligned}
s_{X,\Sigma}(f^\dagger)(x) &= k_X(x)\alpha + p(x)\beta \\
&= [k_X(x) + p(x)\Sigma P_X^\top][K_X + P_X \Sigma P_X^\top]^{-1} f_X^\dagger,
\end{aligned} \tag{A1}$$

$$\begin{aligned}
k_{X,\Sigma}(x, x') &= k(x, x') + p(x)\Sigma p(x')^\top \\
&\quad - [k_X(x) + p(x)\Sigma P_X^\top][K_X + P_X \Sigma P_X^\top]^{-1}[k_X(x') + p(x')\Sigma P_X^\top]^\top
\end{aligned} \tag{A2}$$

*and the coefficients $\alpha$ and $\beta$ are defined via the invertible linear system*

$$\begin{bmatrix} K_X & P_X \\ P_X^\top & -\Sigma^{-1} \end{bmatrix} \begin{bmatrix} \alpha \\ \beta \end{bmatrix} = \begin{bmatrix} f_X^\dagger \\ -\eta \end{bmatrix}. \tag{A3}$$

*Proof.* Under the hierarchical prior we have the marginal

$$f(x) \sim \mathcal{GP}\big(p(x)\eta, k(x, x') + p(x)\Sigma p(x')^\top\big).$$

Thus standard formulae for the conditioning of a Gaussian process [17, Eqns. 2.25, 2.26] can be used:

$$\begin{aligned}
s_{X,\Sigma}(f^\dagger)(x) &= p(x)\eta + [k_X(x) + p(x)\Sigma P_X^\top][K_X + P_X \Sigma P_X^\top]^{-1}[f_X^\dagger - P_X\eta], \\
k_{X,\Sigma}(x, x') &= k(x, x') + p(x)\Sigma p(x')^\top \\
&\quad - [k_X(x) + p(x)\Sigma P_X^\top][K_X + P_X \Sigma P_X^\top]^{-1}[k_X(x') + p(x')\Sigma P_X^\top]^\top.
\end{aligned} \tag{A4}$$

The coefficients $\alpha$ and $\beta$ are therefore

$$\begin{aligned}
\alpha &= [K_X + P_X \Sigma P_X^\top]^{-1}[f_X^\dagger - P_X \Sigma \eta], \\
\beta &= \Sigma\eta + \Sigma P_X^\top[K_X + P_X \Sigma P_X^\top]^{-1}[f_X^\dagger - P_X \Sigma \eta].
\end{aligned}$$

It can be verified by substitution that $P_X^\top \alpha - \Sigma^{-1}\beta = -\eta$ and that the interpolation equations $K_X \alpha + P_X \beta = f_X^\dagger$ hold. This allows us to provide the equivalent characterisation of $\alpha$ and $\beta$ in terms of the linear system in Eqn. A3. To see that this linear system is invertible, we can use the block matrix determinant formula

$$\det\left(\begin{bmatrix} K_X & P_X \\ P_X^\top & -\Sigma^{-1} \end{bmatrix}\right) = \det(-\Sigma^{-1}) \det(K_X + P_X \Sigma P_X^\top).$$

That is, since $\Sigma$ is a positive definite covariance matrix, the block matrix is invertible if and only if $K_X + P_X \Sigma P_X^\top$ is invertible. This is indeed true because, for instance, $K_X + P_X \Sigma P_X^\top$ is the covariance matrix for the random vector $f_X$ under the prior, which is non-singular as $k$ is positive definite and the elements of $X$ are distinct. □

The following Lagrange form [23, Sec. 11.1] of the posterior will be useful:

**Theorem A.1** (Lagrange form for the posterior). *The posterior mean and covariance functions in Eqns. A1 and A2 can be written in the* Lagrange *form*

$$s_{X,\Sigma}(f^\dagger)(x) = u_{X,\Sigma}(x)^\top f_X^\dagger - v_{X,\Sigma}(x)^\top \eta,$$

$$k_{X,\Sigma}(x,x') = k(x,x') + p(x)\Sigma p(x')^\top - [k_X(x) + p(x)\Sigma P_X^\top] u_{X,\Sigma}(x'), \tag{A5}$$

*where*

$$u_{X,\Sigma}(x) := [K_X + P_X \Sigma P_X^\top]^{-1}[k_X(x) + p(x)\Sigma P_X^\top]^\top \tag{A6}$$

*is a vector of* Lagrange cardinal functions *and* $v_{X,\Sigma}(x) := \Sigma[P_X^\top u_{X,\Sigma}(x) - p(x)^\top]$. *These functions are obtained from the invertible linear system*

$$\begin{bmatrix} K_X & P_X \\ P_X^\top & -\Sigma^{-1} \end{bmatrix} \begin{bmatrix} u_{X,\Sigma}(x) \\ v_{X,\Sigma}(x) \end{bmatrix} = \begin{bmatrix} k_X(x)^\top \\ p(x)^\top \end{bmatrix} \tag{A7}$$

*and satisfy the* cardinality property $[u_{X,\Sigma}(x_j)]_i = \delta_{ij}$ *and* $[v_{X,\Sigma}(x_j)]_i = 0$ *for every* $i,j \in \{1,\ldots,n\}$.

*Proof.* From Eqns. A1 and A3, the posterior mean is

$$s_{X,\Sigma}(f^\dagger)(x) = \begin{bmatrix} k_X(x) & p(x) \end{bmatrix} \begin{bmatrix} \alpha \\ \beta \end{bmatrix} = \begin{bmatrix} k_X(x) & p(x) \end{bmatrix} \begin{bmatrix} K_X & P_X \\ P_X^\top & -\Sigma^{-1} \end{bmatrix}^{-1} \begin{bmatrix} f_X^\dagger \\ -\eta \end{bmatrix},$$

and this can be written as $s_{X,\Sigma}(f^\dagger)(x) = u_{X,\Sigma}(x)^\top f_X^\dagger - v_{X,\Sigma}(x)^\top \eta$ where $u_{X,\Sigma}(x)$ and $v_{X,\Sigma}(x)$ are obtained from the linear system in Eqn. A7. The expression for the posterior covariance follows by inserting $u_{X,\Sigma}(x')$, as given in Eqn. A6, into Eqn. A2. The cardinality property follows after we recognise that, setting $x = x_j$, Eqn. A7 is solved by $u_{X,\Sigma}(x_j) = e_j$ (the $j$th unit coordinate vector) and $v_{X,\Sigma}(x_j) = 0$. □

**Theorem 2.5** (Flat prior limit). *Assume that $X$ is a $\pi$-unisolvent set. For the prior in Def. 2.1 we have that $s_{X,\Sigma}(f^\dagger) \to s_X(f^\dagger)$ and $k_{X,\Sigma} \to k_X$ pointwise as $\Sigma^{-1} \to 0$, where*

$$s_X(f^\dagger)(x) = k_X(x)\alpha + p(x)\beta, \tag{A8}$$

$$\begin{aligned} k_X(x,x') = {} & k(x,x') - k_X(x)K_X^{-1}k_X(x')^\top \\ & + \big[k_X(x)K_X^{-1}P_X - p(x)\big]\big[P_X^\top K_X^{-1}P_X\big]^{-1}\big[k_X(x')K_X^{-1}P_X - p(x')\big]^\top, \end{aligned} \tag{A9}$$

*and the coefficients $\alpha$ and $\beta$ are defined via the invertible linear system*

$$\begin{bmatrix} K_X & P_X \\ P_X^\top & 0 \end{bmatrix} \begin{bmatrix} \alpha \\ \beta \end{bmatrix} = \begin{bmatrix} f_X^\dagger \\ -\eta \end{bmatrix}. \tag{A10}$$

*Proof.* For the mean function, the limit can just be taken in the linear system of Eqn. A3 and it is required is to verify that this system can be inverted. From an application of the formula for a block matrix determinant we have that the determinant of the matrix in Eqn. A10 equals $\det(-P_X^\top K_X^{-1}P_X)\det(K_X)$, where $\det(K_X) > 0$. Because $X$ is $\pi$-unisolvent, $P_X$ is of full rank and consequently $\det(-P_X^\top K_X^{-1}P_X) \neq 0$. Thus the block matrix can be inverted.

To obtain the covariance function an additional argument is needed. To this end, the Woodbury matrix identity yields

$$[K_X + P_X \Sigma P_X^\top]^{-1} = K_X^{-1} - K_X^{-1}P_X[\Sigma^{-1} + P_X^\top K_X^{-1}P_X]^{-1}P_X^\top K_X^{-1}.$$

Denoting $L_X := P_X^\top K_X^{-1}P_X$ and inserting the above into Eqn. A4 produces

$$\begin{aligned} k_{X,\Sigma}(x,x') = {} & k(x,x') - k_X(x)K_X^{-1}k_X(x')^\top + p(x)\Sigma p(x')^\top - p(x)\Sigma L_X \Sigma p(x')^\top \\ & - k_X(x)K_X^{-1}P_X \Sigma p(x')^\top - p(x)\Sigma P_X^\top K_X^{-1}k_X(x')^\top \\ & + k_X(x)K_X^{-1}P_X[\Sigma^{-1} + L_X]^{-1}P_X^\top K_X^{-1}k_X(x')^\top \\ & + k_X(x)K_X^{-1}P_X[\Sigma^{-1} + L_X]^{-1}L_X \Sigma p(x')^\top \\ & + p(x)\Sigma L_X[\Sigma^{-1} + L_X]^{-1}P_X^\top K_X^{-1}k_X(x')^\top \\ & + p(x)\Sigma L_X[\Sigma^{-1} + L_X]^{-1}L_X \Sigma p(x')^\top. \end{aligned} \tag{A11}$$

For small enough $\Sigma^{-1}$ we can write the Neumann series
$$[\Sigma^{-1} + L_X]^{-1} = L_X^{-1}\big[I - (L_X\Sigma)^{-1} + (L_X\Sigma)^{-2} - \cdots\big].$$
Therefore we have the trio of results
$$K_X^{-1}P_X[\Sigma^{-1} + L_X]^{-1}P_X^\top K_X^{-1} = K_X^{-1}P_X L_X^{-1}P_X^\top K_X^{-1} + \mathcal{O}(\Sigma^{-1}),$$
$$K_X^{-1}P_X[\Sigma^{-1} + L_X]^{-1}L_X\Sigma = K_X^{-1}P_X\Sigma - K_X^{-1}P_X L_X^{-1} + \mathcal{O}(\Sigma^{-1}),$$
$$\Sigma L_X[\Sigma^{-1} + L_X]^{-1}L_X\Sigma = \Sigma L_X\Sigma - \Sigma + L_X^{-1} + \mathcal{O}(\Sigma^{-1}).$$
Inserting these into Eqn. A11 yields, after cancellation and taking the limit $\Sigma^{-1} \to 0$,
$$k_X(x, x') = k(x, x') - k_X(x)K_X^{-1}k_X(x')^\top$$
$$+ \big[k_X(x)K_X^{-1}P_X - p(x)\big][P_X^\top K_X^{-1}P_X]^{-1}\big[k_X(x')K_X^{-1}P_X - p(x')\big]^\top,$$
as claimed. □

The flat prior limit of the posterior also admits a Lagrange representation:

**Theorem A.2** (Lagrange form in the flat prior limit). *Assume that $X$ is a $\pi$-unisolvent set. The posterior mean and covariance in Eqns. A8 and A9 can be written as*
$$s_X(f^\dagger)(x) = u_X(x)^\top f_X^\dagger - v_X(x)^\top \eta,$$
$$k_X(x, x') = k(x, x') - k_X(x)K_X^{-1}k_X(x')^\top + \big[k_X(x)K_X^{-1}P_X - p(x)\big]v_X(x') \qquad \text{(A12)}$$
*where*
$$v_X(x) := [P_X^\top K_X^{-1}P_X]^{-1}[k_X(x)K_X^{-1}P_X - p(x)]^\top,$$
$$u_X(x) := K_X^{-1}[k_X(x)^\top - P_X v_X(x)]$$
*are obtained from the solution of the invertible linear system*
$$\begin{bmatrix} K_X & P_X \\ P_X^\top & 0 \end{bmatrix}\begin{bmatrix} u_X(x) \\ v_X(x) \end{bmatrix} = \begin{bmatrix} k_X(x)^\top \\ p(x)^\top \end{bmatrix}$$
*and have the cardinality properties $[u_X(x_j)]_i = \delta_{ij}$ and $[v_X(x_j)]_i = 0$ for every $i, j \in \{1, \ldots, n\}$.*

The proof is similar to that of Thm. A.1 and is therefore omitted. Note the reversal in the roles of $u_{X,\Sigma}$ and $v_X$ in Eqns. A5 and A12 for the posterior covariance.

## A.2 Results on Cubature

**Theorem 2.7** (Bayes–Sard cubature; BSC). *Consider the Gaussian process*
$$f(x) \mid \mathcal{D}_X \sim \mathcal{GP}\big(s_{X,\Sigma}(f^\dagger)(x), k_{X,\Sigma}(x, x')\big)$$
*defined in Thm. 2.2 and suppose that $X$ is a $\pi$-unisolvent point set. Then, as $\Sigma^{-1} \to 0$, the mean and variance of $I(f) \mid \mathcal{D}_X$ converge to*
$$\mu_X(f^\dagger) = w_k^\top f_X^\dagger - w_\pi^\top \eta \quad \text{and} \quad \sigma_X^2 = k_{\nu,\nu} - k_{\nu,X}K_X^{-1}k_{\nu,X}^\top + \big(k_{\nu,X}K_X^{-1}P_X - p_\nu\big)w_\pi,$$
*respectively, where the weight vectors $w_k \in \mathbb{R}^n$ and $w_\pi \in \mathbb{R}^Q$ are obtained from the solution of the invertible linear system*
$$\begin{bmatrix} K_X & P_X \\ P_X^\top & 0 \end{bmatrix}\begin{bmatrix} w_k \\ w_\pi \end{bmatrix} = \begin{bmatrix} k_{\nu,X}^\top \\ p_\nu^\top \end{bmatrix}. \qquad \text{(A13)}$$
*Equivalently, $w_k = I(u_X)$ and $w_\pi = I(v_X)$ for the Lagrange functions of Thm. A.2.*

*Proof.* As we have only established that $s_{X,\Sigma}(f^\dagger) \to s_X(f^\dagger)$ and $k_{X,\Sigma} \to k_X$ pointwise in Thm. 2.5, we cannot directly deduce that
$$\mu_{X,\Sigma}(f^\dagger) \to \mu_X(f^\dagger) = \int_D s_X(f^\dagger)(x)\mathrm{d}\nu(x),$$
$$\sigma_{X,\Sigma}^2 \to \sigma_X^2 = \int_D\int_D k_X(x, x')\mathrm{d}\nu(x)\mathrm{d}\nu(x').$$
However, that this is indeed the case can be confirmed by carrying out analysis analogous to that in the proof Thm. 2.5, based on Neumann series, for $\mu_{X,\Sigma}(f^\dagger)$ and $\sigma_{X,\Sigma}^2$ at the limit $\Sigma^{-1} \to 0$. To avoid repetition, the details are omitted. □

**Theorem 2.10.** *Suppose that* $\dim(\pi) = n$ *and let* $X$ *be a* $\pi$*-unisolvent set. If* $\eta = 0$*, then*

$$\mu_X(f^\dagger) = w_k^\top f_X^\dagger, \qquad w_k^\top = p_\nu P_X^{-1}, \qquad \mu_X(p) = I(p) \quad \text{for every } p \in \pi$$

*and*

$$\sigma_X^2 = e_k(X, w_k)^2 = k_{\nu,\nu} - 2k_{X,\nu}w_k + w_k^\top K_X w_k.$$

*That is, the BSC weights* $w_k$ *are the unique weights for a cubature rule with the points* $X$ *such that every function in* $\pi$ *is integrated exactly and the posterior standard deviation* $\sigma_X$ *coincides with the WCE in the RKHS* $H(k)$*.*

*Proof.* Due to $\dim(\pi) = n$ and $X$ being a $\pi$-unisolvent set, the Vandermonde matrix $P_X$ is an invertible square matrix. From Eqn. A13 we have

$$w_k = \left(K_X^{-1} - K_X^{-1}P_X[P_X^\top K_X^{-1}P_X]^{-1}P_X^\top K_X^{-1}\right)k_{\nu,X}^\top + K_X^{-1}P_X[P_X^\top K_X^{-1}P_X]^{-1}p_\nu^\top$$
$$= P_X^{-\top}p_\nu^\top.$$

These are the unique weights satisfying $\sum_{j=1}^n w_{k,j}p_i(x_j) = I(p_i)$ for each basis function $p_i$ of $\pi$. Similarly, the weights $w_\pi$ take the form

$$w_\pi = [P_X^\top K_X^{-1}P_X]^{-1}P_X^\top K_X^{-1}k_{\nu,X}^\top - [P_X^\top K_X^{-1}P_X]^{-1}p_\nu^\top = P_X^{-1}k_{\nu,X}^\top - [P_X^\top K_X^{-1}P_X]^{-1}p_\nu^\top,$$

so that

$$\sigma_X^2 = k_{\nu,\nu} - k_{\nu,X}K_X^{-1}k_{\nu,X}^\top + \left(P_X^\top K_X^{-1}k_{\nu,X}^\top - p_\nu^\top\right)^\top w_\pi$$
$$= k_{\nu,\nu} - k_{\nu,X}K_X^{-1}k_{\nu,X}^\top + \left(P_X^\top K_X^{-1}k_{\nu,X}^\top - p_\nu^\top\right)^\top \left(P_X^{-1}k_{\nu,X}^\top - [P_X^\top K_X^{-1}P_X]^{-1}p_\nu^\top\right)$$
$$= k_{\nu,\nu} - 2k_{\nu,X}w_k + w_k^\top K_X w_k.$$

We recognise this final expression as the squared worst-case error from Eqn. D16. $\qquad \square$

**Corollary 2.11.** *Consider an* $n$*-point cubature rule with points* $X$ *and non-zero weights* $w \in \mathbb{R}^n$ *and assume that* $\nu$ *admits a positive density function (w.r.t. the Lebesgue measure) and that* $\eta = 0$*. Then there is a function space* $\pi$ *of dimension* $n$*, such that the BSC method recovers* $w_k = w$ *and* $\sigma_X^2 = e_k(X, w)^2$*, as defined in Thm. 2.7.*

*Proof.* From the assumption that $\nu$ has a positive density function with respect to the Lebesgue measure it follows that $\nu(\{x\}) = 0$ for every $x \in D$ and that for any distinct points $x_1, \ldots, x_n \in D$ there exist disjoint sets $D_i$ of positive measure such that $x_i \in D_i$. Select then the $n$ functions

$$p_i = \mathbb{1}_{D_i \setminus \{x_i\}} + \frac{\nu(D_i)}{w_i}\mathbb{1}_{\{x_i\}}.$$

It holds that $I(p_i) = \nu(D_i)$. The associated Vandermonde matrix is diagonal and has the elements $[P_X]_{ii} = \nu(D_i)/w_i$. Hence it can be trivially inverted. It follows that the BSC method with basis $\{p_1, \ldots, p_n\}$ has a posterior mean $\mu_X(f^\dagger) = w^\top f_X^\dagger$. $\qquad \square$

The construction is more appealing if the weights are positive and their sum does not exceed one, since then we can use $p_i = \mathbb{1}_{D_i}$ for disjoint sets such that $\nu(D_i) = w_i$ and $x_i \in D_i$, or if the weights are naturally given by exactness conditions on $\pi$ and $X$ is $\pi$-unisolvent. Examples of such more natural constructions include uniformly weighted (quasi) Monte Carlo rules, that arise from using a partition $D = \cup_{i=1}^n D_i$ with $\nu(D_i) = 1/n$, and Gaussian tensor product rules.

# B   Unisolvent Point Sets

This section contains more details and examples about unisolvent point sets.

**Definition 2.4** (Unisolvency). Let $\pi$ denote a finite-dimensional linear subspace of real-valued functions on $D$. A point set $X = \{x_1, \ldots, x_n\} \subset D$ with $n \geq \dim(\pi)$ is called $\pi$-*unisolvent* if the zero function is the only element in $\pi$ that vanishes on $X$.

The following proposition provides an equivalent operational characterisation of unisolvency:

**Proposition B.1.** *Let $\{p_1, \ldots, p_Q\}$ denote a basis of $\pi$, so that $Q = \dim(\pi)$. Then a point set $X$ is $\pi$-unisolvent if and only if the $n \times Q$ Vandermonde matrix $P_X$ is of full rank.*

**Example B.2** (Cartesian product of a unisolvent set). As a simple example of how one can generate a unisolvent set in $\mathbb{R}^d$, consider the Cartesian grid $X = Z^d$ for a set $Z = \{z_1, \ldots, z_m\} \subset \mathbb{R}$ of distinct points. Then for any $d$-variate polynomial

$$p \in \Pi := \mathrm{span}\{x^\alpha \colon \alpha \in \mathbb{N}_0^d,\ \alpha_1, \ldots, \alpha_d \le m - 1\},$$

the univariate polynomial

$$p_j(z) = p(z_{\alpha_1}, \ldots, z_{\alpha_{j-1}}, z, z_{\alpha_{j+1}}, \ldots, z_{\alpha_d})$$

is of degree at most $m - 1$ and, for any indices $j \in \{1, \ldots, d\}$ and $\alpha_1, \ldots, \alpha_d \in \{1, \ldots, m-1\}$, the polynomial $p_j$ cannot vanish on $Z$ unless it is the zero polynomial. It follows that $p$ cannot vanish on $X$ unless $p \equiv 0$. Therefore $X$ is $\Pi$-unisolvent. Note that $\#X = \dim(\Pi) = m^d$.

**Example B.3** (Not all sets are unisolvent). As a counterexample, consider six points $X = \{(x_i, y_i), i = 1, \ldots, 6\}$ on a unit circle in $\mathbb{R}^2$. These points are not $\Pi_2(\mathbb{R}^d)$-unisolvent (polynomials of degree at most two; see Eqn. C14): the associated Vandermonde matrix

$$P_X = \begin{bmatrix} 1 & x_1 & y_1 & x_1 y_1 & x_1^2 & y_1^2 \\ 1 & x_2 & y_2 & x_2 y_2 & x_2^2 & y_2^2 \\ 1 & x_3 & y_3 & x_3 y_3 & x_3^2 & y_3^2 \\ 1 & x_4 & y_4 & x_4 y_4 & x_4^2 & y_4^2 \\ 1 & x_5 & y_5 & x_5 y_5 & x_5^2 & y_5^2 \\ 1 & x_6 & y_6 & x_6 y_6 & x_6^2 & y_6^2 \end{bmatrix}$$

for the canonical polynomial basis is not of full rank as the first column is the sum of the last two columns.

Intuitively, "almost all" point sets are unisolvent, but to actually verify that an arbitrary point set $X$ is unisolvent, from Proposition B.1 it is required to compute the rank of the Vandermonde matrix $P_X$, which entails a super-linear computational cost [21]. However, certain point sets are guaranteed to be unisolvent:

- When $\pi$ is a Chebyshev system (so that its basis functions are so-called *generalised polynomials*) in one dimension, any set $X \subset \mathbb{R}$ of distinct points is $\pi$-unisolvent [10].

- For $\pi$ spanned by the indicator functions $\mathbb{1}_{A_1}, \ldots, \mathbb{1}_{A_n}$ of disjoint sets $A_i \subset D$ such that $x_i \in A_i$, the set $X$ is $\pi$-unisolvent and $P_X$ is the $n \times n$ identity matrix.

- *Padua points* on $[-1, 1]^2$ are known to be unisolvent with respect to polynomial spaces [4].

- Recent algorithms for generating moderate number of points for polynomial interpolation, with a unisolvency guarantee on the output, can be used [21, 7].

## C  Convergence Results

This section contains fundamental convergence results for the cubature rule $\mu_X$ associated with the mean of the BSC output.

### C.1  Rates of Convergence for Gaussian and Sobolev Kernels

For standard BC, the analogous convergence results can be found in [3, 9]. Our attention is restricted to the case when $\pi$ is the space $\Pi_m(\mathbb{R}^d)$ of $d$-variate polynomials of degree at most $m \ge 0$:

$$\Pi_m(\mathbb{R}) := \mathrm{span}\{x^\alpha \colon \alpha \in \mathbb{N}_0^d,\ \alpha_1 + \cdots + \alpha_d \le m\}, \tag{C14}$$

where $\alpha$ is a multi-index and $x^\alpha = x_1^{\alpha_1} \cdots x_d^{\alpha_d}$. It is noteworthy that Thm. C.4 has been derived, essentially in the form we present it, in non-probabilistic setting already in [2]. However, we go beyond [2] and provide convergence results for both the Gaussian kernel, as well as kernels of the Matérn class.

To establish convergence, we observe that the posterior mean $s_X(f^\dagger)$ defined in Eqn. A8 coincides with the interpolant defined in [23, Sec. 8.5] for a conditionally positive definite kernel[1]. The extensive convergence theory outlined in [23, Ch. 11] can be therefore brought to bear. For a set $X \subset D$ and $D$ bounded, define the *fill distance* $h_{X,D} := \sup_{x \in D} \min_{i=1,\dots,n} \|x - x_i\|$. Considered as a sequence of sets indexed by $n \in \mathbb{N}$, we say $X$ is *quasi-uniform* in $D$ if $h_{X,D} \lesssim n^{-1/d}$, where $a_n \lesssim b_n$ is used to signify that the ratio $a_n/b_n$ is bounded above for sufficiently large $n \in \mathbb{N}$. Recall also that $\|\cdot\|_k$ is the norm of the RKHS $H(k)$ induced by the kernel $k$.

**Theorem C.1** (Spectral convergence for Gaussian kernels). *Let $D$ be a hypercube in $\mathbb{R}^d$, let $\nu$ admit a density which is bounded, let $X$ be a $\Pi_m(\mathbb{R}^d)$-unisolvent set for some $m \geq 0$, and let $k$ be a Gaussian kernel: $k(x, x') = \exp(-\|x - x'\|^2/(2\ell^2))$ for some $\ell > 0$. Then there is a $c > 0$ such that, for a quasi-uniform point set,*

$$|\mu_X(f^\dagger) - I(f^\dagger)| \lesssim e^{-(c/d)n^{1/d} \log n} \|f^\dagger\|_k.$$

*Proof.* That there is a $h_0 > 0$ such that $\|s_X(f^\dagger) - f^\dagger\|_\infty \leq e^{c \log(h_{X,D})/h_{X,D}} \|f^\dagger\|_k$ whenever $h_{X,D} < h_0$ was established in [23, Thm. 11.22]. The remainder of the proof is transparent. $\qquad\square$

The next result extends [9, Prop. 4] for the standard BC method. Its proof follows that of Thm. C.1 and is an application of [23, Cor. 11.33]. The following two notions are needed for stating this result.

**Definition C.2** (Interior cone condition). A bounded domain $D \subset \mathbb{R}^d$ is said to satisfy an *interior cone condition* if there exists an angle $\theta \in (0, \frac{\pi}{2})$ and a radius $r > 0$ such that for each $x \in D$ a unit vector $\xi(x)$ exists such that the cone $\{x + \lambda y : y \in \mathbb{R}^d, \|y\|_2 = 1, y^\top \xi(x) \geq \cos\theta, \lambda \in [0, r]\}$ is contained in $D$.

The interior cone condition rules out domains that contain "pinch points" on their boundary. This means that the domains we use in the numerical examples, $\mathbb{R}^d$ and hypercubes, satisfy an interior cone condition.

**Definition C.3** (Norm-equivalence). Two norms $\|\cdot\|_1$ and $\|\cdot\|_2$ on a space $V$ are said to be *equivalent* if there exist finite positive constants $C_1$ and $C_2$ such that $C_1 \|v\|_1 \leq \|v\|_2 \leq C_2 \|v\|_1$ for every $v \in V$.

Furthermore, in what follows the boundary of the domain $D$ is required to be *Lipschitz*; see [9, Sec. 3]. This essentially means that the boundary cannot be "too irregular". Indeed, most domains of interest to us, such as convex sets, have a boundary that is Lipschitz.

**Theorem C.4** (Polynomial convergence for Sobolev kernels). *Let $X$ be a $\Pi_m(\mathbb{R}^d)$-unisolvent set for some $m \geq 0$. Suppose that (i) $D$ is a bounded open set that satisfies an interior cone condition and whose boundary is Lipschitz; (ii) for $\alpha > d/2$, the RKHS of the kernel $k$ is norm-equivalent to the standard Sobolev space $H^\alpha(D)$ and (iii) $\nu$ admits a density function that is bounded. Then, for a quasi-uniform point set,*

$$|\mu_X(f^\dagger) - I(f^\dagger)| \lesssim n^{-\alpha/d} \|f^\dagger\|_{H^\alpha(D)}.$$

**Remark C.5.** The Matérn kernel

$$k(x, x') = \frac{2^{1-\rho}}{\Gamma(\rho)} \left( \frac{\sqrt{2\rho}\, \|x - x'\|}{\ell} \right)^\rho K_\rho \left( \frac{\sqrt{2\rho}\, \|x - x'\|}{\ell} \right), \tag{C15}$$

where $K_\nu$ is the modified Bessel function of the second kind, with length-scale $\ell > 0$ and smoothness parameter $\rho > 0$ satisfies Assumption (ii) of the above theorem with $\alpha = \rho + d/2$.

## C.2 Explicit Rates of Convergence (for the case $Q = n$)

As pointed out in Cor. 2.11, the mean $\mu_X(f^\dagger)$ of the Bayes–Sard output can be arranged to coincide with any given cubature rule through judicious choice of the function space $\pi$, provided that its dimension matches the number of nodes $x_i$ that are used. In this case, convergence rates are trivially inherited. For example, and for simplicity letting $\nu$ be uniform on $D = [0, 1]^d$,

- nodes drawn randomly (or through utilisation of a Markov chain) from $\nu$ and uniform weights yield the standard (probabilistic) Monte Carlo rate

$$\mathbb{E}\big(\big|\mu_X^{\mathrm{MC}}(f^\dagger) - I(f^\dagger)\big|^2\big)^{\frac{1}{2}} \lesssim n^{-1/2}\|f^\dagger\|_{L^2(D)};$$

- certain quasi-Monte Carlo methods can attain polynomial rates for functions in the space $H_{\mathrm{mix}}^\alpha(D)$ of dominating mixed smoothness:

$$\big|\mu_X^{\mathrm{QMC}}(f^\dagger) - I(f^\dagger)\big| \lesssim n^{-\alpha+\varepsilon}\|f^\dagger\|_{H_{\mathrm{mix}}^\alpha(D)}$$

for any $\varepsilon > 0$. See [6, Ch. 15] for these results and for the formal definition of the norm;

- certain sparse grid methods on hypercubes have the rates

$$\big|\mu_X^{\mathrm{SG}}(f^\dagger) - I(f^\dagger)\big| \lesssim n^{-\alpha/d}(\log n)^{(d-1)(\alpha/d+1)}\|f^\dagger\|_{C^\alpha(D)},$$
$$\big|\mu_X^{\mathrm{SG}}(f^\dagger) - I(f^\dagger)\big| \lesssim n^{-\alpha}(\log n)^{(d-1)(\alpha+1)}\|f^\dagger\|_{F^\alpha(D)}$$

for function having bounded derivatives or bounded mixed derivatives up to order $\alpha$, respectively. See [13, 14, 15] for these results and for formal definitions of the norms.

## D    An Equivalent Kernel Perspective

In this section we interpret the output of the BSC method from the perspective of the reproducing kernel, in order to provide additional insight that complements the main text. The formulation of cubature rules in reproducing kernel Hilbert spaces dates back to [11, 18, 20, 19, 12] and in particular the integrated kernel interpolant was studied in [2] and [22].

### D.1    Interpolation

There is a well-understood equivalence between Gaussian process regression and optimal interpolation in reproducing kernel Hilbert spaces: Let $\{p_1, \ldots, p_Q\}$ be a basis for $\pi$ and define the kernel $k_\pi(x, x') = \sum_{i=1}^Q p_i(x)p_i(x')$. Consider the kernel

$$k_\sigma(x, x') = k(x, x') + \sigma^2 k_\pi(x, x')$$

for $\sigma > 0$. Then the reproducing kernel Hilbert space induced by $k_\sigma$ corresponds to the set

$$H(k_\sigma) = \big\{f + p : f \in H(k),\, p \in \pi\big\}$$

equipped with a particular $\sigma$-dependent inner product. It can be shown that the interpolant of $\mathcal{D}_X$ with minimal norm in $H(k_\sigma)$ is unique and given by

$$s_{X,\sigma}(f^\dagger)(x) = [k_X(x) + \sigma^2 k_{\pi,X}(x)][K_X + \sigma^2 P_X P_X^\top]^{-1} f_X^\dagger,$$

where the row vector $k_{\pi,X}(x)$ has the elements $k_\pi(x, x_j)$. When $\eta = 0$, it is straightforward to show that $s_{X,\sigma}(f^\dagger) = s_{X,\Sigma}(f^\dagger)$ for $\Sigma = \sigma^2 I$ and thus $s_{X,\sigma}(f^\dagger) \to s_X(f^\dagger)$ pointwise as $\sigma \to \infty$. The kernel interpolation operator $s_X$ is well-studied and the reader is referred to, for example, Sec. 8.5 of [23].

### D.2    Cubature

The worst-case error $e_k(X, w)$ of a cubature rule described by the points $X = \{x_1, \ldots, x_n\} \subset D$ and weights $w = (w_1, \ldots, w_n) \in \mathbb{R}^n$ has the explicit form

$$e_k(X, w) := \sup_{\|h\|_k \leq 1} \left|\sum_{i=1}^n w_i h(x_i) - \int_D h\,\mathrm{d}\nu\right| = \big(k_{\nu,\nu} - 2k_{\nu,X}w + w^\top K_X w\big)^{1/2}. \qquad \text{(D16)}$$

See for example [16, Cor. 3.6].

Recall from Sec. 2.4 that the weights $w_{\mathrm{BC}}$ of the standard BC method are worst-case optimal in the reproducing kernel Hilbert space $H(k)$ induced by the kernel $k$:

$$w_{\mathrm{BC}} = \arg\min_{w \in \mathbb{R}^n} e_k(X, w).$$

Conveniently, the minimum corresponds to the integration error for the kernel mean function $k_\nu$ which acts as the representer of integration (i.e., $\langle h, k_\nu \rangle_k = I(h)$ for $h \in H(k)$):

$$e_k(X, w_{\mathrm{BC}}) = \left(k_{\nu,\nu} - k_{\nu,X} w_{\mathrm{BC}}\right)^{1/2}.$$

Now, turning to BSC, we have from Sec. D.1 that the BSC rule $\mu_X(f^\dagger)$ can be cast as an optimal cubature method based on the kernel

$$k_\sigma(x, x') = k(x, x') + \sigma^2 k_\pi(x, x')$$

in the $\sigma \to \infty$ limit. The following therefore holds for the weights $w_k$ and variance $\sigma_X^2$ of the BSC method:

$$w_k = \lim_{\sigma \to \infty} \underset{w \in \mathbb{R}^n}{\arg\min}\, e_{k_\sigma}(X, w), \qquad \sigma_X^2 = \lim_{\sigma \to \infty} \min_{w \in \mathbb{R}^n} e_{k_\sigma}(X, w)^2.$$

Recall that $H(k_\sigma)$ consists of functions which can be expressed as sums of elements of $H(k)$ and $\pi$. To simplify the following argument, assume $f^\dagger \in H(k_\sigma)$. That the elements of $\pi$ are exactly integrated can be clearly understood in this context. Indeed, the norm of a function $h \in H(k_\sigma)$ is [1, Sec. 4.1]

$$\|h\|_{k_\sigma}^2 = \min_{g \in H(k),\, p \in \pi} \left\{ \|g\|_k^2 + \sigma^{-2}\|p\|_{k_\pi}^2 : g + p = h \right\},$$

where we have used the fact that scaling a kernel by $\sigma^2$ results in scaling the RKHS inner product by $\sigma^{-2}$. Thus $\|\sigma^2 p\|_{k_\sigma}^2 \leq 1$ for any $p \in \pi$ such that $\|p\|_{k_\pi}^2 \leq 1$. Consequently, the worst-case error (D16) is dominated by the integration error for functions in $\pi$:

$$e_{k_\sigma}(X, w) = \sup_{\|h\|_{k_\sigma} \leq 1} \left| \sum_{i=1}^n w_i h(x_i) - \int_D h \, \mathrm{d}\nu \right| \geq \sigma^2 \sup_{\|p\|_{k_\pi} \leq 1} \left| \sum_{i=1}^n w_i p(x_i) - \int_D p \, \mathrm{d}\nu \right|.$$

It follows that the BSC rule must be the unique cubature rule that integrates exactly functions from $\pi$, existence of which is guaranteed by the $\pi$-unisolvency assumption on $X$. In particular, when $\dim(\pi) = n$, the weights $w_k$ are fully-determined by the requirement of exactness for functions in $\pi$ and nothing is done to integrate functions in $H(k)$ well. Consequently, the limiting variance $\sigma_X^2$ must coincide with the (squared) worst-case error $e_k(X, w_k)^2$ in the RKHS $H(k)$.

**Remark D.1.** Alternatively, the limiting weights $w_k$ can be seen as a solution to the constrained convex optimisation problem of minimising the RKHS approximation error to the kernel mean function $k_\nu$ under exactness conditions for functions in $\pi$:

$$w_k = \underset{w \in \mathbb{R}^n}{\arg\min}\, \|k_\nu - k_X w\|_k \quad \text{subject to} \quad P_X^\top w = p_\nu^\top.$$

This can be verified in a straightforward manner based on [5, Sec. 5.2].

## E  Further Details for Numerical Experiments

This section contains further details about the zero coupon bonds example of Sec. 3.3. See [8, Sec. 6.1] for a complete account.

The $d_T$-step Euler–Maruyama discretisation with uniform step-size $\Delta t = T/d_T$ of the Vasicek model

$$\mathrm{d}r(t) = \kappa\big(\theta - r(t)\big)\mathrm{d}t + \sigma \mathrm{d}W(t), \tag{E17}$$

where $W(t)$ is the standard Brownian motion and $\kappa$, $\theta$, and $\sigma$ are positive parameters, is

$$r_{t_i} = r_{t_{i-1}} + \kappa\big(\theta - r_{t_{i-1}}\big)\Delta t + \sigma\sqrt{\Delta t}x_{t_i}, \quad i = 1, \ldots, d,$$

for independent standard Gaussian random variables $x_{t_i}$ and some deterministic initial value $r_{t_0}$. The quantity of interest is the Gaussian expectation

$$P(0, T) := \mathbb{E}\left[ \exp\left( -\Delta t \sum_{i=0}^{d_T-1} r_{t_i} \right) \right] = \exp(-\Delta t r_{t_0})\mathbb{E}\left[ \exp\left( -\Delta t \sum_{i=1}^{d_T-1} r_{t_i} \right) \right]$$

of dimension $d = d_T - 1$. This expectation admits the closed-form solution

$$P(0, T) = \exp\left( -\frac{(\gamma + \beta_{d_T} r_{t_0})T}{d} \right)$$

for certain constants $\gamma$ and $\beta_{d_T}$. In the experiment, we used the same parameter values as in [8] for the model in Eqn. E17:

$$\kappa = 0.1817303, \quad \theta = 0.0825398957, \quad \sigma = 0.0125901, \quad r_{t_0} = 0.021673, \quad T = 5.$$

## Footnotes

[1] Note that a positive definite kernel is also a conditionally positive definite kernel.