[Reviews · NeurIPS 2018]

Reviewer 1



The proposes a novel approximation for (possibly high dimensional) integrals. As in Bayesian quadrature, the value of the integral is estimated by integrating over a Gaussian process trained on the currently known function values. The main contribution of this paper is the combination of Bayesian integration (e.g., integrate over a nonparametric GP) with parametric methods (e.g. integrate over a polynomial as classical quadrature does). With this approach, the benefits of classical numerical integration can be combined with the merits of a fully Bayesian approach. The paper also includes convergence analysis and derives several well known classical rules as a special case of the proposed method. In my opinion, this is a great paper. The problem addressed is one of the core problems in machine learning to date -- approximate intractable posteriors, which is synonymous to approximate intractable integrals. The paper is well written, technically sound and gives a nice insight on Bayesian quadrature and it's relation to classical quadrature methods. The proposed method is evaluated on a one dimensional toy example and one high dimensional example, where a closed form solution is available. minor comments: -in the zero coupon bonds experiment , the relative error does not decrease monotonically with increasing n, as opposed to BC and MC. Please explain this effect. -It would be nice to see BSC applied to some machine learning quadrature problem (e.g. approximate a posterior) and conduct a comparison to othe approximation methods, e.g., variational approaches.

Reviewer 2



Summary: This works proposes a version of the Bayesian cubature method. Essentially, a Gaussian process model is fitted to the data and to approximate the original integral. Thus the integral of the Gaussian process is the estimate of the original targeted integration. Strength: The technical description, definition, proofs and remarks are rigorous and clear. Weakness: The main idea of the paper is not original. The entire Section 2.1 is classical results in Gaussian process modeling. There are many papers and books described it. I only point out one such source, Chapter 3 and 4 of Santner, Thomas J., Brian J. Williams, and William I. Notz. The design and analysis of computer experiments. Springer Science & Business Media, 2013. The proposed Bayes-Sard framework (Theorem 2.7), which I suspected already exist in the Monte Carlo community, is a trivial application of the Gaussian process model in the numerical integration approximation. The convergence results, Theorem 2.11 and Theorem 2.12, are also some trivial extension of the classic results of RKHS methods. See Theorem 11.11 and 11.13 of Wendland, Holger. Scattered data approximation. Vol. 17. Cambridge university press, 2004. Or Theorem 14.5 of Fasshauer, Gregory E. Meshfree approximation methods with MATLAB. Vol. 6. World Scientific, 2007. Quality of this paper is relatively low, even though the clarity of the technical part is good. This work lacks basic originality, as I pointed out in its weakness. Overall, this paper has little significance.

Reviewer 3



Post rebuttal: thank you for your answers. It would be interesting to see comparison to Clenshaw-Curtis cubature method in the camera ready version and I hope the future work will explore the quality of the BSC uncertainty estimates. I'd also be excited to see applications to problems such as posterior mean estimation. This paper proposes a new Bayesian numerical integration approach. Taking the Bayesian viewpoint allows not only to estimate the integral of interest, but also quantify the estimation uncertainty. On the downside, previously known Bayesian Cubature approach may be inaccurate in the integral estimation in high dimensions. Proposed Bayes-Sard Cubature (BSC) extends Bayesian Cubature (BC) by endowing the mean function of a GP with a (flat) prior over linear subspace of a user-specified family of functions (e.g., polynomials). As a result, under some conditions, BSC can exactly recover elements of this subspace, allowing it to mimic behavior of non-Bayesian cubature approaches, while maintaining non-degenerate uncertainty quantification. I think the paper is written well and it is easy to follow the logic of the results (even though I am not an expert in the topic and unfamiliar with many of the cited works). Ability to equip any cubature method with uncertainty quantification seems useful and novel contribution. Proposed method is studied thoroughly from the theoretical viewpoint and provided intuition for the technical results along with explicit references to related literature make it an enjoyable read. GP with the prior on the mean function could be an interesting modeling approach for other problems. I have several questions: - When you state that any cubature rule can be recovered in the BSC framework, does it imply that there is always a tractable way of setting the corresponding functions space or it only implies existence of such? - You showed that BSC is less sensitive than BC to length-scale misspecification, however it would be interesting to see how sensitive is BSC to this parameter when compared against non-Bayesian approaches. Will it require fine-tuning of the parameter to achieve same performance (in terms of estimation error)? - You mentioned that quality of the uncertainty quantification is a matter of future studies, however I think some empirical comparison to deterministic cubatures could make the submission stronger. Hopefully BSC can match the integral estimation error and show higher uncertainty in some meaningful examples when the estimates of baseline methods are inaccurate. Do you have any insight in this regard? - I found it interesting that, in Fig. 3, BC always performs quite well for l=sqrt(d), however for BSC the relative performance for different length-scale choices tends to switch around. Can you explain the reason for it?

Reviewer 4



The paper develops a new integration method which follows from a Gaussian process (GP) modeling of the integrand. The author starts by defining the posterior of the GP model, which appears as an approximation of the integrand. Fixing some parameters in the GP models, the author recovers the standard Bayesian cubature method (rk 2.3). The originality of their proposal, the Bayes-Sard cubature (BSC) method, comes from considering a limiting case in the GP model (which characterizes the most uninformative prior). The estimate given by the BSC method is the integral of the mean of the resulting posterior Gaussian process. It is a linear integration rule and another nice property is that it integrates perfectly the functions lying in some linear space pi (chosen by the user). The paper is reasonably organized and the ideas are well explained (with good examples). On top of that the paper surveys nicely many results by putting in light new links between some existing methods. Here are some points/questions that might help to improve the paper: - The main contribution of the paper seems more methodological than theoretical but too few experiments are given in the end to validate the approach. - The proposed method depends on a linear space pi (a notation which I find a bit exotic for a linear subspace). It is not clearly stated that the integrals of functions in pi should be known for the method to work. This being said, there might be some link with the control variates method as detailed for instance in the archived paper arXiv:1801.01797. In section 2.4 of the previous reference, the authors obtain a formulation with cubature weights of control variates (which integrates perfectly the functions in a linear space that correspond to your pi). In the present paper, the author gives some weights that, as well, are integrating perfectly the functions in pi. In Th 2.9 the weights are claimed to be unique. Does it means that both methods coincide? (the statement on uniqueness might require more details). - The statements of the theorems are too vague and it is sometimes difficult to understand their meaning. For instance, ‘’in the limit’’ is not clear while a standard equation would do the job. Especially, in Th 2.7, where one cannot know what means ‘’in the limit’’ without checking the proof. Another example is ‘’consider an n points cubature rule’’. This has not been defined. - Even if interesting, the convergence analysis is not the very point of the paper (as it follows from known results). It would be preferable to furnish some remarks on the rate and perhaps to compare with other methods (in place of theoretical consideration that would perfectly fit the supplementary material). - Section 2.1 results from classical techniques from the GP literature. It should be mentioned from the beginning (for the non expert reader). - page 3. line 105. As the meaning of the remark coincides with the definition of s and k (line 97), it seems to me redundant. - page 3. line 112. The remark should be at the beginning of section 2.1.3. - remark 2.13. It is said that the supplementary material ‘’elaborates’’ on assumptions (i) and (ii) whereas it just gives some definitions. I was disappointing to see no further explanation on the assumptions.